# Modeling the Health Impact and Cost-Effectiveness of a Combined Schoolgirl HPV Vaccination and Cervical Cancer Screening Program in Guangdong Province, China

**DOI:** 10.3390/children11010103

**Published:** 2024-01-15

**Authors:** Yating Huang, Dantao Zhang, Lihua Yin, Jianguo Zhao, Zhifeng Li, Jing Lu, Xiaoming Zhang, Chenggang Wu, Wei Wu

**Affiliations:** 1School of Public Health, Guangdong Pharmaceutical University, Guangzhou 510200, China; 2Guangdong Provincial Institute of Public Health, Guangdong Provincial Center for Disease Control and Prevention, Guangzhou 511430, China; 3The Second Division Center for Disease Control and Prevention of Xinjiang Production and Construction Corps, Tiemenguan 841007, China; 4Guangdong Provincial Center for Disease Control and Prevention, Guangzhou 511430, China

**Keywords:** human papillomavirus vaccination, cervical cancer, elimination, screening, modeling, cost-effectiveness, economic evaluation

## Abstract

Low human papillomavirus (HPV) vaccine uptake is a key barrier to cervical cancer elimination. We aimed to evaluate the health impact and cost-effectiveness of introducing different HPV vaccines into immunization programs and scaling up the screening program in Guangdong. We used a dynamic compartmental model to estimate the impact of intervention strategies during 2023–2100. We implemented the incremental cost-effectiveness ratio (ICER) in costs per averted disability-adjusted life year (DALY) as an indicator to assess the effectiveness of the intervention. We used an age-standardized incidence of 4 cases per 100,000 women as the threshold for the elimination of cervical cancer. Compared with the status quo, scaling up cervical cancer screening coverage alone would prevent 215,000 (95% CI: 205,000 to 227,000) cervical cancer cases and 49,000 (95% CI: 48,000 to 52,000) deaths during 2023–2100. If the coverage of vaccination reached 90%, domestic two-dose 2vHPV vaccination would be more cost-effective than single-dose and two-dose 9vHPV vaccination. If Guangdong introduced domestic two-dose 2vHPV vaccination at 90% coverage for schoolgirls from 2023 and increased the screening coverage, cervical cancer would be eliminated by 2049 (95% CI 2047 to 2051). Introducing two doses of domestic 2vHPV vaccination for schoolgirls and expanding cervical cancer screening is estimated to be highly cost-effective to accelerate the elimination of cervical cancer in Guangdong.

## 1. Introduction

Cervical cancer is a major public health problem worldwide, with an estimated 604,000 new cases, and 342,000 deaths reported in 2020, mainly affecting low- and middle-income countries (LMICs) [1,2]. Human papillomavirus (HPV) vaccination and screening are effective ways to prevent cervical cancer [3]. Despite considerable efforts, screening coverage remains low in many LMICs, and the global vaccine coverage is still suboptimal, with 70% of girls worldwide living in countries that had not yet introduced the HPV vaccine by the end of 2019 [4]. High prices and inadequate supply are the major barriers to HPV vaccine introduction, resulting in growing disparities between countries [5,6,7]. 

In 2020, cervical cancer was the second-most common cancer type in women in China, with an estimated 109,700 new cases and more than 59,000 deaths [8]. HPV vaccination and screening are effective ways to reduce the incidence and mortality of cervical cancer [3]. In fact, China has not yet included HPV vaccination in its national immunization program (NIP) [9]. The limited supply of HPV vaccine is sold privately and is mostly administered to women aged 15–45 using a three-dose schedule, leaving the World Health Organization (WHO)-recommended primary target group of 9–14-year-old females largely unvaccinated. 

Intending to accelerate the elimination of cervical cancer, the Chinese government has developed an expansion of screening coverage plan [10], targeting a screening coverage of 50% in women aged 35–64 by 2025, with the goal of reaching 70% by 2030. Meanwhile, Guangdong province has included the vaccination of adolescent girls with two doses of the domestic HPV vaccine before sexual debut in 2023 [11]. Therefore, we have an unprecedented opportunity to establish a universal vaccination program for schoolgirls, together with an expansion of cervical cancer screening for adult women, to achieve the goal of eliminating cervical cancer.

Five types of HPV vaccines (two domestically manufactured 2vHPVs (Cecolin^®^ and Walrinvax^®^, Yuxi, China) as well as imported 2vHPV (Cervarix^®^), 4vHPV (Gardasil^®^), and 9vHPV vaccines (Gardasil-9^®^)) are currently available in China. The first domestically manufactured HPV vaccine (Cecolin^®^) was approved in China in December 2019 and pre-qualified by the WHO in October 2021. The second domestically produced HPV vaccine (Walrinvax^®^) was also approved in China in 2022. Walrinvax^®^ is currently undergoing WHO prequalification review and is anticipated to be involved in global use shortly afterward. Notably, the latest research evidence indicated that domestic 2vHPV vaccine is comparable in immunogenicity to the imported 2vHPV vaccine [12,13,14,15]. Meanwhile, there are 17 HPV vaccines under clinical trials, with nine of them in phase III, including five 9vHPV vaccines. These developments signal that China’s domestically developed HPV vaccines are on the verge of market introduction. Consequently, this progress is expected to provide the population with a wider range of vaccination options [6,16]. Furthermore, there is growing evidence that a single-dose vaccine could provide high and sustained protection against HPV infection and related diseases [17]. In 2022, the WHO’s Strategic Advisory Group of Experts on Immunization (SAGE) reviewed the evidence on single-dose HPV vaccination and recommended reduced dose schedules. One or two doses are recommended for females aged 9–20 years, while females aged 21 years or older should receive two doses with a 6-month interval. This is considered a potential measure for expediting progress towards the goal of vaccinating 90% of girls by age 15 by 2030 [18,19]. Therefore, to alleviate the constraint of HPV vaccines, it is necessary to estimate the health and economic impact of single-dose and two-dose 9vHPV vaccination strategies. 

In this study, we developed an age-structured compartmental model based on local parameters to evaluate the health impact and cost-effectiveness of introducing different HPV vaccines into immunization programs and scaling up the screening in Guangdong province.

## 2. Methods

### 2.1. Model Structure

An age-structured compartmental model was used to represent the HPV disease progression through the transmission cycle, including elements such as population dynamics, vaccination, screening, and treatment programs. The HPV transmission portion was adapted from the work of Xiaomeng Ma et al., 2020 [20] and the age structure came from an otherwise unrelated disease transmission model [21]. Herein, the pre-vaccine steady state was used as the initial value of the model, and the model was solved to obtain the full-time path to finally bring the system to a steady state. Appendix A depicts the general structure of the transmission and age compartments of the proposed model.

### 2.2. Inputs and Assumptions

The parameters representing the natural history of HPV infection were synthesized from previous modeling studies [13,22,23,24,25,26,27,28,29,30,31,32,33,34] (Appendix A).Those related to the acceptability of cervical screening, screening sensitivity, and cure rates of different treatment methods were specifically obtained from peer-reviewed literature [35,36,37] (Appendix A). The data on HPV prevalence used to calibrate the model were obtained from epidemiological studies [38,39]. The actual observed cervical cancer incidence and mortality rates between 2010 and 2016 were extracted from the Guangdong cancer registries (Appendix A). Demographic data, fertility rate, and age-specific all-cause mortality rates were obtained from the Guangdong Statistical Yearbook 2019 (http://stats.gd.gov.cn/, accessed on 1 June 2022). The costs of domestically manufactured 2vHPV vaccine and imported 9vHPV vaccine were USD 46.63 and USD 191.73 per dose, respectively [40,41], and the vaccine administration cost was estimated at USD 3.83 per dose. The screening costs and treatment costs (CIN1/CIN2/CIN3/cervical cancer) were extracted from previous modeling studies [40,42,43] (Appendix A). These costs were converted from Chinese yuan into US dollars (in 2023).

### 2.3. Model Calibration 

We calibrated the model by referring to the epidemiological data of HPV prevalence, incidence, and mortality rates of cervical cancer observed in Guangdong. We applied the Latin hypercube sampling method to randomly pick up parametric values from their confidence interval and generated 2000 different parametric value sets. The mean least squared error was selected as the criterion for calibration. We evaluated the accuracy of outcomes based on 2000 parametric value sets and selected the top 1% results to conduct the subsequent intervention projection. All calibration processes were conducted on the whole population adhering to the status quo strategy and the assumption that the demographic characteristic remained unchanged. 

### 2.4. Alternative Strategies

Given budget constraints, our model proposed a school-based HPV vaccination strategy for girls aged 13–14 years [16]. On the basis of the previous modeling studies [14,44,45], for the domestic 2vHPV vaccine, we assumed its cross-protection equivalent to the 4vHPV vaccine due to their shared aluminum adjuvant formulation. Therefore, we considered only the domestic 2vHPV (Cecolin^®^, Xiamen, China) and 9vHPV (Gardasil^®^9, Whitehouse Station, NJ, USA) vaccines independently in our study. The status quo was assumed to be the current vaccination state (no vaccination) and the current coverage of the 3-yearly cervical cancer screening program was around 31.8% in women aged 35–64 years [46]. 

According to the Chinese accelerated elimination of cervical cancer action plan, screening coverage should reach 50% in women aged 35–64 years by 2025, and escalate to 70% by 2030 [10]. Therefore, the target population for cervical cancer screening was women aged 35–64 years, with screening initiation at 35 due to the peak prevalence of cervical issues. Our model simplifies screening into primary and secondary screenings. For the primary screening, women aged 35–64 years are recommended to conduct cervical cancer screening using HPV DNA test and cytological tests. Secondary screening, using colposcopy or biopsy, confirms the results of primary screening. Women showed positive signs in primary screening are advised to undergo secondary screening for confirmation. In the scale-up screening intervention strategy, we assumed that screening would switch to HPV DNA-based testing in 2023, with the screening coverage increasing beyond 70% by 2030. The alternative strategies evaluated in the study are listed in Table 1.

### 2.5. Vaccine Efficacy

Based on the totality of empirical comparative evidence for the efficacy and effectiveness of a single dose [47,48], we assumed that the duration of protection and efficacy provided by a single-dose vaccine would be inferior to that of a 2-dose vaccine. A 2-dose regimen was assumed to provide lifetime protection [6,49,50]. Given that there was no evidence of waning in vaccine protection after a single dose over more than 10 years of follow-up in the Indian and CVT studies [17,51], we conservatively assumed a 30-year duration of protection for single-dose strategies [49]. Herein, for cervical cancer, the degree of 2-dose 2vHPV vaccination protection was given as 0.70, and the degree of 2-dose 9vHPV vaccination protection was set as 0.90 [50].

### 2.6. Main Outcome

First, the cumulative cervical cancer cases, deaths, and DALYs in each intervention strategy were reported for 2023–2100. The intervention effectiveness is represented by DALYs. By taking the burden from multiple HPV induced morbidities into account, DALYs measure the gap between current health status and ideal health status of the entire population. DALYs are composed of years of life lost (YLL) due to premature mortality in the population, and the years lost due to disability (YLD). Second, we considered the Segi standardized world population in calculating the age-standardized incidence. Cervical cancer elimination was defined as the first year in which the age-standardized incidence rate falls below 4 per 100,000 women [52,53]. We used the currency exchange rate in 2023 and the annual price discount rate of 3% to estimate the costs in US dollars (USD), and took the gross domestic product (GDP) per capita (USD 13,956) as the ICER threshold [54]. Finally, we examined the cost-effectiveness of the strategies and then selected the optimal strategies considered with the year of cervical cancer elimination. Given the latency period of several decades for the development of cervical cancer after infection, a time horizon of 77 years was chosen to fully assess the effectiveness of different strategies. This 77-year timeframe is consistent with the goal of eliminating cervical cancer, as it allows a comprehensive assessment of the dynamic factors contributing to its eradication in present and future generations.

### 2.7. Sensitivity Analyses

We performed one-way sensitivity analyses to explore changes in the assumed sensitivity of different tests (±10%), the domestic 2vHPV vaccine efficacy (Cecolin^®^) (±10%), the domestic 2vHPV vaccine cost per dose (±10%), the treatment cost (±10%), and the assumed discount rate for cost (1% and 5%). The threshold analyses were carried out according to the results of tornado diagrams. Meanwhile, a probability sensitivity analysis was also performed on 9vHPV vaccine cost fluctuation and uncertainties around prevalence variations. A Latin hypercubic sampling method was adopted to randomly extract values from price ranges and epidemiological parameter ranges to construct parametric matrices for simulation. In total, 1000 random scenarios were generated to visualize the uncertainty. The age-structured compartmental model and subsequent intervention projection were both conducted using R 4.1.0 (R Core Team, https://www.R-project.org/ (accessed on 1 June 2022), version 4.1.0, R Foundation for Statistical Computing).

## 3. Results

### 3.1. Model Calibration

Figure 1 illustrates that the calibration on HPV prevalence, incidence, and the mortality rate of cervical cancer were in good agreement with the observed local epidemiological outcomes.

### 3.2. Public Health Impact of Strategies

Our model indicated that under the status quo (3-year screening coverage of 31.8%), the age-standardized cervical cancer incidence rate in Guangdong Province would increase from 9.64/100,000 to 11.07/100,000 by 2100. In contrast, scaling up cervical cancer screening coverage alone would prevent 215,000 (95% CI: 205,000 to 227,000) cervical cancer cases and 49,000 (95% CI: 48,000 to 52,000) deaths for the period of 2023–2100. Compared to the status quo, scaling-up cervical cancer screening alone would be cost-effective, with a discounted ICER of USD 1130 (95% CI: 1073 to 1230) per DALY (Table 2). The future scaling up the screening revealed a huge impact on cervical cancer prevention in these scenarios, while increasing the screening coverage alone was insufficient to eliminate cervical cancer (Table 3 and Figure 2).

### 3.3. Cost-Effectiveness of Strategies

The cost-effectiveness of each strategy was considered using a standard ICER-based decision rule to determine the cost-effectiveness frontier. Ordered from least effective at the top to most effective at the bottom, the net costs, DALYs and ICER for each strategy are shown in Table 2. Overall, compared with the status quo, all vaccine introduction combined with different screening strategies were cost-effective (Table 2 and Appendix A). In the context of 31.8% screening, compared with no vaccination, two-dose 2vHPV vaccination would be more costly and effective, with the benefits increasing as the coverage expands. If the coverage of vaccination reached 90%, compared to two-dose 2vHPV vaccination, single-dose 9vHPV vaccination and two-dose 9vHPV vaccination would have an ICER of −USD 6263 (95% CI: −5217 to −6667)/DALY and USD 42,750 (95% CI: 38,000 to 57,000)/DALY, and thus would not be on the cost-effectiveness frontier (Table 2).

In addition, if the screening coverage increased beyond 70% by 2030, the cost-effectiveness results would still hold, but the ICERs would somewhat increase (Table 2). Once again, when the coverage of vaccination was up to 90%, compared to two-dose 2vHPV vaccination, single and two-dose 9vHPV vaccination would be weakly dominated, with −USD 11,182 (95% CI: −8786 to −11,280)/DALY and USD 68,400 (95% CI: 57,000 to 68,200)/DALY. Meanwhile, compared to single-dose vaccination, two-dose 9vHPV would have an ICER of USD 13,687 (95% CI: 10,950 to13,750)/DALY. Therefore, immediate two-dose 2vHPV vaccination for girls aged 13–14 in 2023 and the rapid scale-up of screening to achieve 70% coverage in 2030 is the most effective alternative and thus dominates all others. If Guangdong Province adopts the most effective strategy, the annual age-standardized incidence of cervical cancer is predicted to decrease to fewer than four new cases per 100,000 women in Guangdong by 2049 (95% CI 2047 to 2051) (Table 2 and Figure 2).

### 3.4. Sensitivity Analysis

Figure 3 depicted the results of the one-way sensitivity analysis. None of the sensitivity analyses changed our fundamental finding. The model simulation indicated that the discount rate exerted a moderate effect on the main outcome measure while changing the sensitivity of different tests. Sensitivity of screening, the cost of 2vHPV vaccination, and the cost of treatment had the least measurable effect. Probabilistic sensitivity analyses are displayed in scatter plots for discounted incremental costs and DALYs under intervening strategies with different 9vHPV vaccine prices (Figure 4). This indicated that the switch to 9vHPV would not be considered cost-effective in China without a decrease in the current market price of the vaccine.

## 4. Discussion

To our knowledge, this is the first instance when a dynamic model has been used to simulate the cost-effectiveness analysis of simulating HPV vaccination strategies for girls aged 13–14 years combined with scale-up screening in Guangdong Province from both governmental and societal perspectives. The model was calibrated to HPV prevalence, incidence and mortality rate of cervical cancer, and the results were in good agreement with the observed local epidemiological outcomes.

Our study demonstrates that the age-standardized incidence of cervical cancer in Guangdong will increase substantially between 2023 and 2100 if the current vaccination and screening coverage is maintained. Importantly, even if the screening target is achieved in 2030, Guangdong would not be able to achieve cervical cancer elimination by the end of the century if the universal vaccination program for schoolgirls had not been initiated. This aligns with a previous study emphasizing that screening alone is not sufficient to eliminate cervical cancer [20,45]. There are several reasons for the inadequacy of screening alone. First, cervical cancer screening only detects HPV infection only after it has occurred and therefore does not prevent the initial acquisition of the virus. In contrast, HPV vaccination not only provides immunity before sexual debut but also reduces the risk of acquiring HPV during sexual activity, significantly lowering the incidence of HPV-related cancers later in life. Second, cervical cancer screening coverage varies widely among people from different socioeconomic backgrounds; people from resource-limited settings are less likely to accept regular screening. Hence, a universal vaccination program for schoolgirls can provide relatively equitable access to HPV prevention for disadvantaged populations and compensate for inequalities in access to cervical cancer screening.

Furthermore, we found that the domestic two-dose 2vHPV vaccination strategy initiated in 2023 would be more cost-effective than no vaccination. If Guangdong introduces domestic two-dose 2vHPV vaccination at 90% coverage for girls aged 13–14 years from 2023 onwards and the screening target is achieved in 2030, we are likely to achieve the elimination of cervical cancer by the 2050s. Numerous published studies have shown that the introduction of 2vHPV vaccines in China is highly cost-effective [44,55,56]. Our study adds to the cumulative evidence that a combined intervention approach of the vaccination of schoolgirls with an expanded cervical cancer screening program will lead to a significant reduction in new infections and cervical cancer-related mortality. At the same time, the domestically produced 2vHPV vaccine is easier to scale up and has lower manufacturing costs than imported vaccines, making it potentially suitable for inclusion in both local and NIP in China [13].

We also found that irrespective of the screening coverage, the single-dose 9vHPV vaccination is likely to be more cost-effective compared to the two-dose 9vHPV vaccination under pessimistic assumptions of one-dose 30-year duration of protection, which is generally consistent with previous modeling studies [50,57,58]. Evidence from immunogenicity trials and post-randomization studies has indicated that a single vaccine dose of vaccine may offer comparable long-lasting protection against infection and clinical endpoints as two or three doses [6,17]. Routine single-dose HPV vaccination will release vaccine doses, enabling a faster expansion of the vaccination scale, and thus achieving the maximum health and economic benefits within the constrained supply in China [58]. In the meantime, it is worth noting that China’s pricing for the 9vHPV vaccine is beyond the reach of most eligible women and government-funded programs, prompting future research related to optimal vaccine pricing and price negotiations with manufacturers. Therefore, the approval process for domestically produced HPV vaccines should be accelerated, especially for new-generation vaccines targeting more types of HPV, in order to force manufacturers to effectively reduce prices.

Our study has several strengths. First, it is the first of its kind to incorporate an dynamic transmission model to assess the health and economic impact of large-scale HPV vaccination and screening programs on cervical cancer elimination in Guangdong Province. Second, it provides timely evidence to inform the current and future HPV vaccination strategy planning in Guangdong. Third, we considered several policy-relevant outcomes, including the population-level effectiveness, cost-effectiveness, and time for cervical cancer elimination, to provide comprehensive evidence for policymakers in Guangdong.

Limitations of this analysis include the exclusion of HPV genotypes due to data complexity; the omission of administration, logistics, and storage the costs for HPV vaccination; and the assumption of a constant population size and a fixed age distribution throughout the simulation. Meanwhile, we did not explore male vaccination strategies and focused on the primary target of young girls, as recommended by WHO’s Strategic Advisory Group of Experts on Immunization (SAGE). Lastly, our analysis did not examine other diseases caused by HPV, such as oropharyngeal, vagina, head, and neck cancers, etc., and thus may have underestimated the effects of this vaccine [59]. Our study assumes that screening intensity will remain constant in the future, with efforts focused on increasing screening coverage. Under this assumption, the results suggest that the 2vHPV vaccine is more cost-effective than 9vHPV. However, studies in Norway have shown that a reduction in screening intensity makes the 9vHPV vaccine more cost-effective at current prices [60]. Therefore, it is necessary to include changes in screening intensity in the model to analyze the impact of increased or decreased screening intensity on the cost-effectiveness of the 2vHPV or 9vHPV vaccines in the future. 

## 5. Conclusions

Overall, our study shows that the incidence of cervical cancer in Guangdong Province will continue to increase under the current prevention strategies. Under pessimistic assumptions of 30 years of single-dose protection, routine single-dose vaccination could avert most of the cervical cancers averted by two-dose vaccination, while being more efficient, easier to implement and less costly. Furthermore, the introduction of two doses of domestic 2vHPV vaccination for schoolgirls and the expansion of cervical cancer screening is predicted to be a highly cost-effective method of accelerating cervical cancer elimination in Guangdong with important implications for cervical cancer control in China and other resource-limited countries.

## Figures and Tables

**Figure 1 children-11-00103-f001:**
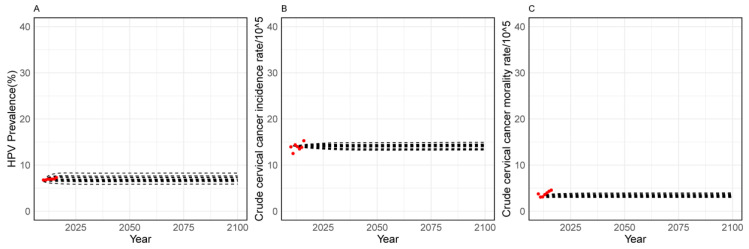
Conditions versus model output after calibration. (**A**–**C**) represent the following variables: HPV prevalence (%), Crude cervical cancer incidence rate (/10^5^), Crude cervical cancer mortality rate (/10^5^); Red dot represents observed local epidemiological values, black dot indicates predicted values from the model.

**Figure 2 children-11-00103-f002:**
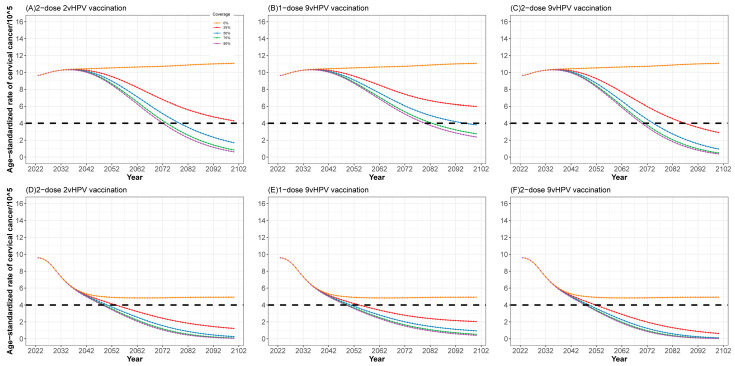
Projection of age-standardized incidence of cervical cancer with different strategies between 2023 and 2100. (**A**–**C**) at the current screening rate; (**D**–**F**) in the context of expanding cervical cancer screening.

**Figure 3 children-11-00103-f003:**
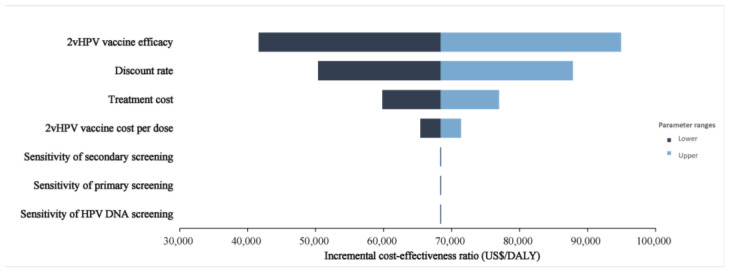
Tornado diagram for two-dose 2vHPV vaccination versus two-dose 9vHPV vaccination ICER in the context of expanding cervical cancer screening.

**Figure 4 children-11-00103-f004:**
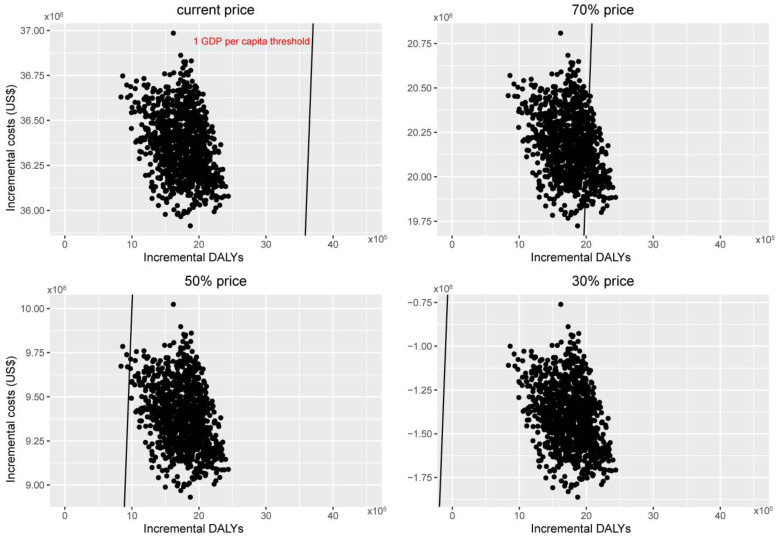
Two-dimensional cost-effectiveness plane demonstrating the distribution of 1000 simulations: two-dose 2vHPV vaccination versus two-dose 9vHPV vaccination in the context of expanding cervical cancer screening at different (100%, 70%, 50%, 30%) price level.

**Table 1 children-11-00103-t001:** Alternative strategies evaluated in this study.

Strategies	Screening	Vaccination
1: Status quo	31.8% 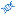	No vaccination
2		25%, 50%, 75%, 90% 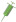 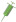
3		25%, 50%, 75%, 90% 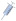
4		25%, 50%, 75%, 90% 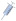 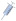
5	71.8% 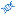 by 2030, by 5% increments per year	No vaccination
6		25%, 50%, 75%, 90% 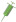 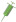
7		25%, 50%, 75%, 90% 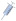
8		25%, 50%, 75%, 90% 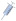 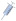

Notes: 
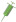
 2vHPV vaccination at 13–14 years of age; 
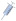
 9vHPV vaccination at 13–14 years of age; 
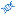
 Screening at 35–64 of age every 3 years.

**Table 2 children-11-00103-t002:** ICER of strategies that lie on the efficiency frontier compared with the next-best strategy in 2023–2100.

Vaccination Strategy	Coverage (%)	Current Screening Rate			Expanding Screening Rate		
		Discount Cost (USD) 10^7^	Discounted DALYs 10^4^	3% Discounted ICER	Discount Cost (USD) 10^7^	Discount DALYs 10^4^	3% Discounted ICER
No vaccination	0	421 (410, 435)	434 (378, 485)	/	534 (517, 553)	334 (291, 375)	/
2-dose 2vHPV	25	456 (448, 467)	361 (316, 403)	479 (390, 613)	540 (528, 554)	270 (238, 305)	94 (14, 204)
	50	479 (472, 489)	330 (289, 367)	742 (611, 888)	552 (543, 566)	246 (216, 276)	500 (143, 682)
	75	497 (491, 506)	315 (276, 350)	1267 (1059, 1462)	567 (559, 581)	236 (207, 264)	1500 (1250, 1778)
	90	507 (500, 515)	309 (271, 344)	1667 (1422, 1800)	575 (568, 589)	233 (204, 260)	2116 (2000, 3000)
1-dose 9vHPV	25	526 (518, 538)	375 (329, 419)	/	616 (603, 631)	281 (247, 317)	/
	50	579 (572, 590)	348 (306, 389)	1963 (1733, 2348)	658 (647, 672)	259 (228, 292)	1910 (1640, 2316)
	75	612 (605, 622)	335 (294, 374)	2538 (2133, 2750)	686 (676, 699)	248 (219, 279)	2545 (2077, 3222)
	90	626 (620, 636)	329 (289, 367)	2333 (2000, 3000)	698 (689, 712)	244 (215, 274)	3000 (2600, 3250)
2-dose 9vHPV	25	642 (634, 652)	347 (304, 387)	/	720 (710, 735)	259 (228, 291)	/
	50	752 (745, 761)	318 (279, 354)	3793 (3303, 4440)	822 (814, 837)	239 (209, 267)	5100 (4250, 5473)
	75	820 (813, 829)	306 (268, 340)	5667 (4857, 6182)	888 (881, 902)	231 (202, 257)	8250 (6500, 9571)
	90	849 (842, 858)	302 (265, 335)	7250 (5800, 9667)	917 (909, 930)	228 (199, 254)	9667 (9337, 11,106)

**Table 3 children-11-00103-t003:** Accumulated cervical cancer cases and deaths of strategies in 2023–2100.

Vaccination Strategy	Coverage (%)	Current Screening Rate		Expanding Screening Rate	
		Accumulated Cervical Cancer Cases 10^3^	Accumulated Deaths 10^3^	Accumulated Cervical Cancer Cases 10^3^	Accumulated Deaths 10^3^
No vaccination	0	534 (514, 545)	119 (113, 142)	319 (287, 341)	70 (64, 89)
2-dose 2vHPV	25	391 (378, 407)	93 (85, 110)	219 (203, 233)	51 (47, 64)
	50	333 (323, 345)	80 (73, 94)	185 (174, 194)	44 (40, 54)
	75	306 (296, 319)	74 (67, 87)	172 (163, 180)	42 (38, 50)
	90	297 (287, 309)	72 (76, 84)	168 (160, 175)	41 (38, 50)
1-dose 9vHPV	25	419 (401, 435)	98 (90, 117)	237 (218, 254)	54 (50, 68)
	50	367 (356, 383)	88 (80, 103)	204 (190, 217)	48 (44, 60)
	75	341 (335, 355)	80 (75, 94)	190 (178, 201)	44 (41, 56)
	90	331 (326, 346)	78 (73, 91)	184 (173, 195)	41 (40, 54)
2-dose 9vHPV	25	360 (351, 376)	87 (79, 102)	203 (189, 214)	47 (43, 59)
	50	311 (300, 323)	75 (68, 88)	174 (165, 183)	42 (38, 51)
	75	290 (280, 303)	70 (64, 82)	166 (158, 173)	40 (37, 49)
	90	283 (273, 296)	68 (63, 80)	163 (155, 169)	39 (36, 48)

## Data Availability

The data presented in this study are available in article and Appendix A.

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
