# Peer review of "Modeling the Health Impact and Cost-Effectiveness of a Combined Schoolgirl HPV Vaccination and Cervical Cancer Screening Program in Guangdong Province, China"

_children, 2024, doi:10.3390/children11010103_

Round 1

Reviewer 1 Report

Comments and Suggestions for Authors

Dear Authors,

Your study is extremely important, as HPV vaccination was implemented in many countries.

I believe your manuscript must be published, but some modification are needed.

Introduction should be improved.

-Please explain more about domestic HPV vaccines.

-Please detail the HPV vaccination algorithm, I am not familiar with one doze of vaccine, WHO recommends Two doses up to the age of 15 years old, I could not find the one dose recomandation.

The methods are fine and well presented.

Results section must  be improved.

Tables 2 and 3 must be redone, they are not very clear, too many numbers.

Discussion section is well written.

Author Response

I would like to take this opportunity to express my sincere gratitude for your time and efforts in reviewing my manuscript. Your positive feedback and encouragement have been instrumental in enhancing the quality of the final manuscript, and I am truly honored to have had the benefit of your expertise and insights.

To facilitate the review process, we have provided a point-by-point response to each of the comments.

Please find below our point-by-point response to your comments:

Reviewer 1

Introduction should be improved.

-Please explain more about domestic HPV vaccines.

-Please detail the HPV vaccination algorithm, I am not familiar with one dose of vaccine, WHO recommends Two doses up to the age of 15 years old, but I could not find the one dose recommendation. WHO One-dose Human

Response: Thanks for your valuable suggestion. We have added to the manuscript. (lines 65-78, lines 78-80)

The first domestically manufactured HPV vaccine (Cecolin®) was approved in China in December 2019 and pre-qualified by WHO in October 2021. The second domestically produced HPV vaccine (Walrinvax™) was also approved in China in 2022. Walrinvax® is currently undergoing the WHO prequalification review and is anticipated to be involved for global use shortly afterward. Notably, the latest research evidence indicated that the domestic 2vHPV vaccine is comparable in immunogenicity to the imported 2vHPV vaccine. Meanwhile, there are 17 HPV vaccines under clinical trials, with 9 of them in phase III, including 5 9vHPV vaccines. These developments signal that China's domestically developed HPV vaccines are on the verge of market introduction. Consequently, this progress is expected to provide the population with a wider range of vaccination options.

- In 2022, WHO's Strategic Advisory Group of Experts on Immunisation (SAGE) reviewed the evidence on single-dose HPV vaccination and recommended reduced dose schedules. One or two doses are recommended for females aged 9-20 years, while females aged 21 years or older should receive two doses at a 6-month interval. This is considered a potential measure to expedite progress towards the goal of vaccinating 90% of girls by the age of 15 by 2030.

Results section must be improved. Tables 2 and 3 must be redone, they are not very clear, too many numbers.

Response: We have redone Tables 2 and 3, trying to make them more visual. As the results section will utilize data from each column, we cannot remove any columns. We hope you will be satisfied with the modified outcome.

We tried our best to improve our manuscript and made some changes in the manuscript. We appreciate the reviewer’s warm work earnestly and hope that the correction will meet with approval. Once again, thank you very much for your comments and suggestions.

Looking forward to hearing from you soon.

Yours Sincerely,

               on behalf of all authors.

Reviewer 2 Report

Comments and Suggestions for Authors

Primary prevention through HPV vaccination represents the basis of cervical cancer control. The authors embark on a modeling study implementing models and undertaking assumptions which are shared with other similar state of the art modeling studies.

Disappointingly, it is unclear which of the two domestic bivalent vaccines (Cecolin or Walrinvax) has been implemented in the study. Referring simply to “2vHPV (bivalent)” vaccine is confusing and potentially misleading; perhaps it would be more appropriate that vaccine's interventions are quoted with their brand names. Furthermore, since most HPV vaccination modeling studies are funded by the pharmaceutical industry, a clear statement regarding potential COI’s of all authors would be of great value.

Line 83: The quoted price for Gardasil9 is unrealistic; much more competitive pricing can be achieved through tendering especially for central governments.

Line 84: Have the authors incorporated Self Sampling (Vaginal/Urine) approaches in their screening strategies assumptions?

Lines 99-100: “Because the 2vHPV and 4vHPV vaccines have the same protective effect for cervical cancer” Again, it is unclear which bivalent vaccine are the authors referring to. If this is the GSK Cervarix, then there is huge literature (irrelevant with this submission) supporting the notion that protective effects are SIMILAR (but not the same) because of the ASO4 adjuvant of Cervarix and cross-reactivity of Gardasil-4.

Lines 112-113: Again, which 2vHPV vaccine? Long term efficacy data for domestic Chinese bivalent vaccines are lacking and cannot be extrapolated projecting from other VLP’s. The authors need to substantiate with several references their assumption for a 30 or 50(!!!) yrs vaccination protection.

Lines 128: Which of the two domestic 2vHPV vaccines?

Line 208: (Table 2) “Deaths” and not “Death Cases”.

Line 256: “China's pricing for the 9vHPV vaccine is beyond the reach of most eligible women” please refer to comment as in line 83.

Two more comments: 1) both bivalent Cecolin and Walrinvax are currently only licensed for girls. Why don’t the author’s consider a modeling study with gender neutral vaccination (GNV) which currently represents the most effective strategy to tackle HPV16 & the HPV-related oropharyngeal Cancer pandemic? 2) Type replacement of HPV genotypes targeted by the current VLP vaccines has already alarming effects (Pimenoff et al, Cell Host & Microbe2023); how can the authors’ ambitious strategy spanning 25 years disregard this trend and rely on a bivalent vaccine solely targeting HPV16&18?

Comments on the Quality of English Language

Minor language editing might be required.

Author Response

I would like to take this opportunity to express my sincere gratitude for your time and efforts in reviewing my manuscript. Your positive feedback and encouragement have been instrumental in enhancing the quality of the final manuscript, and I am truly honored to have had the benefit of your expertise and insights.

To facilitate the review process, we have provided a point-by-point response to each of the comments.

Please find below our point-by-point response to your comments:

Primary prevention through HPV vaccination represents the basis of cervical cancer control. The authors embark on a modeling study implementing models and undertaking assumptions which are shared with other similar state of the art modeling studies.

Disappointingly, it is unclear which of the two domestic bivalent vaccines (Cecolin or Walrinvax) has been implemented in the study. Referring simply to “2vHPV (bivalent)” vaccine is confusing and potentially misleading; perhaps it would be more appropriate that vaccine's interventions are quoted with their brand names. Furthermore, since most HPV vaccination modeling studies are funded by the pharmaceutical industry, a clear statement regarding potential COI’s of all authors would be of great value.

Response: We apologize for not making this clear in the original manuscript. we only considered domestic 2vHPV (Cecolin®) and 9vHPV (Gardasil®9) independently in our study. We have added it in our manuscript. Meanwhile, the authors declare that the research was conducted in the absence of any commercial or financial relationships that could be construed as a potential conflict of interest. We hope this clarifies the issue for you. Thanks for your valuable suggestion. (lines 136-138, lines 398-400)

Line 83: The quoted price for Gardasil9 is unrealistic; much more competitive pricing can be achieved through tendering especially for central governments.

Response: We agree with your suggestion. In the strategies in which a government-sponsored schoolgirl HPV vaccination program is to be launched, the price of vaccination would probably be further negotiated down. Additionally, with future increases in production capacity for domestic vaccines and the development of a domestic low-cost 9vHPV vaccine, vaccine shortages will likely ease, and vaccine prices may decrease further, especially if vaccines can be purchased as a part of a national tender. We therefore assumed a decline in this parameter in the sensitivity analyses. (lines 194-198, lines 249-253, Figure 4)

Line 84: Have the authors incorporated Self Sampling (Vaginal/Urine) approaches in their screening strategies assumptions?

Response: Our study does not consider the inclusion of self-sampling (vaginal/urine) approaches in the screening strategies. Self-sampling (vaginal/urine) approaches have shown promising results in cervical cancer screening, both as a standalone primary screening method and when combined with liquid-based cytology (LBC) for triage screening. Self-sampling helps overcome many barriers to screening, increases participation among high-risk women, especially when done at home, and eliminates the reliance on clinic hours, transportation, and language barriers. It is a convenient and relatively comfortable method. However, the limited availability and high cost of self-sampling kits, along with low awareness among women about self-sampling methods, present challenges in promoting the use of self-sampling for HPV testing and expanding screening coverage in China. Additionally, the Chinese government primarily relies on clinician-collected sampling for large-scale cervical cancer screening. Furthermore, due to the lack of available data on self-sampling rates, our study does not consider the inclusion of self-sampling (vaginal/urine) approaches in the screening strategies. Thanks for your valuable suggestion.

Lines 99-100: “Because the 2vHPV and 4vHPV vaccines have the same protective effect for cervical cancer” Again, it is unclear which bivalent vaccine are the authors referring to. If this is the GSK Cervarix, then there is huge literature (irrelevant with this submission) supporting the notion that protective effects are SIMILAR (but not the same) because of the ASO4 adjuvant of Cervarix and cross-reactivity of Gardasil-4.

Response: We apologize for not making this clear in the original manuscript. We only considered domestic 2vHPV (Cecolin®) and 9vHPV (Gardasil®9) independently in our study. Considering the inaccurate expression of this sentence, we have revised it as“Based on the previous modeling studies, for the domestic 2vHPV vaccine, we assumed its cross-protection equivalent to the 4vHPV vaccine due to their shared aluminum adjuvant formulation.” (lines 132-133)

Lines 112-113: Again, which 2vHPV vaccine? Long term efficacy data for domestic Chinese bivalent vaccines are lacking and cannot be extrapolated projecting from other VLP’s. The authors need to substantiate with several references their assumption for a 30 or 50(!!!) yrs vaccination protection.

Response: We apologize for not making this clear in the original manuscript. We only considered domestic 2vHPV (Cecolin®) and 9vHPV (Gardasil®9) independently in our study. The 5.5-year follow-up suggests that the domestically produced 2vHPV vaccine (Cecolin®) shows no waning of protection after vaccination (Zhao FH et al, Lancet Infect Dis). Meanwhile, many Chinese scholars in the field of modeling have assumed a duration of lifetime for the two-dose vaccination strategy (Tingting You et al, EClinical Medicine; Zhuoru Zou MM et al, Lancet Glob Health)). Therefore, a 2-dose regimen was assumed to provide lifetime protection in our study. The 30 years of average protection were chosen as pessimistic scenarios given that there was no evidence of waning in vaccine protection after one dose through more than 10 years of follow-up in the Indian and CVT studies (Partha Basu et al, Lancet Public Health; Kreimer AR et al, J Natl Cancer Inst). Meanwhile, I followed the modeling study (Bénard É, Drolet M, et al, Lancet Public Health), assuming a 30-year duration of protection.

Lines 128: Which of the two domestic 2vHPV vaccines?

Response: We apologize for not making this clear in the original manuscript. We only considered domestic 2vHPV (Cecolin®) and 9vHPV (Gardasil®9) independently in our study. We have revised it.

Line 208: (Table 2) “Deaths” and not “Death Cases”.

Response: Thanks for your valuable comments. We have revised it.

Line 256: “China's pricing for the 9vHPV vaccine is beyond the reach of most eligible women” please refer to comment as in line 83.

Response: Thanks for your valuable comments. We assumed a decline in this parameter in the sensitivity analyses. (Figure 4)

Two more comments: 1) both bivalent Cecolin and Walrinvax are currently only licensed for girls. Why don’t the author’s consider a modeling study with gender neutral vaccination (GNV) which currently represents the most effective strategy to tackle HPV16 & the HPV-related oropharyngeal Cancer pandemic? 2) Type replacement of HPV genotypes targeted by the current VLP vaccines has already alarming effects (Pimenoff et al, Cell Host & Microbe2023); how can the authors’ ambitious strategy spanning 25 years disregard this trend and rely on a bivalent vaccine solely targeting HPV16&18?

Response 1: We agree with your suggestion. it might be beneficial to explore the potential of a gender-neutral vaccination strategy. Such an approach would not only protect boys from HPV-related cancers but also enhance herd protection for girls who are not vaccinated, thereby reducing the overall prevalence of HPV in the population. We also know that countries such as the United States, Austria, and Canada which are supported by the Global Alliance for Vaccines and Immunization have been offering HPV vaccine to both genders for several years.

However, because of the lack of local data, our analysis could not examine other diseases caused by HPV, such as oropharyngeal, vagina, head and neck cancers. Meanwhile, according to WHO's global strategy for cervical cancer elimination, the primary target is vaccinating young girls who could directly benefit from it. WHO's Strategic Advisory Group of Experts on Immunisation (SAGE) has recommended that vaccinating boys should be delayed until current vaccine supply constraints are alleviated. Furthermore, there is currently no nationwide policy in China that mandates or recommends HPV vaccination for boys. Therefore, we have added your suggestion to our limitation in the manuscript. we believe that further analysis is necessary. Therefore, we plan to continue our analysis in the future. We appreciate your feedback and will consider it as we further develop our study. (lines 360-362)

Response 2: Thank you for your suggestion. I have read the article and found that if China provides free vaccination using the current VLP vaccines, it could lead to alarming effects. However, because the Chinese government has not yet introduced this vaccine, it will take a long time from its introduction to free mass vaccination. Meanwhile, taking into consideration the price and supply of HPV vaccines and the preference of Chinese authorities, China would likely introduce domestically manufactured HPV vaccines into the NIP. The timeline for Chinese companies to independently develop and widely apply vaccines is lengthy. Additionally, I have reviewed articles by Chinese scholars and found no mention of incorporating this vaccine into their model. Therefore, it seems unlikely to be included in this study's consideration for mainland China's free vaccination strategy in the foreseeable future.

Minor language editing might be required.

Response: We truly agree with your suggestion. We understand the importance of language proficiency in academic writing. We are pleased to inform you that the manuscript has already been sent to a colleague fluent in English writing to improve the language quality. We hope that this will improve the readability and overall quality of the manuscript for your evaluation.

We tried our best to improve our manuscript and made some changes in the manuscript. We appreciate the reviewer’s warm work earnestly and hope that the correction will meet with approval. Once again, thank you very much for your comments and suggestions.

Looking forward to hearing from you soon.

Yours Sincerely,

               on behalf of all authors.

Reviewer 3 Report

Comments and Suggestions for Authors

Huang et al. have investigated the health impact and cost-effectiveness of integrating human papillomavirus (HPV) vaccinations and enhanced cervical cancer screening in Guangdong Province, China. They utilized a dynamic compartmental model spanning from 2023 to 2100 to assess different intervention strategies. Their evaluation centered on the incremental cost-effectiveness ratio (ICER) in terms of costs per averted disability-adjusted life-year (DALY). The study found that increasing cervical cancer screening alone could prevent approximately 215,000 cancer cases and 49,000 deaths within the specified period. Notably, a 90% coverage with the domestic two-dose 2vHPV vaccine emerged as more cost-effective compared to the single-dose and two-dose 9vHPV vaccines. Implementing this vaccine for schoolgirls, along with increased screening coverage, could lead to the elimination of cervical cancer in Guangdong by 2049. Overall, the research suggests that a combined approach of introducing the two-dose 2vHPV vaccine and expanding screening programs is a highly cost-effective strategy to accelerate cervical cancer elimination in the region.

The claims are properly placed in the context of the previous literature. The experimental data support the claims. The manuscript is written clearly enough that most of it is understandable to non-specialists. The authors have provided adequate proof for their claims, without overselling them. The authors have treated the previous literature fairly. The paper offers enough details of methodology so that the experiments could be reproduced.

Comments

1. Firstly, it appears that your analysis is primarily focused on evaluating the cost-effectiveness of various HPV vaccines, differing vaccine dosages, vaccination coverage levels, and screening coverage rates. While this is undoubtedly important, I noticed that your model does not seem to include the costs associated with the screening program itself, particularly in a vaccinated cohort. The financial implications of screening are critical, especially considering that screening costs can significantly surpass those of HPV vaccination. This is due to the recurring nature of screening throughout an individual's life, in contrast to the one-time or two-time administration of the HPV vaccine.

Additionally, the logistical and economic impacts of screening on adults should be factored into your analysis. Unlike schoolchildren who can be vaccinated during school hours without disrupting work schedules, adults typically need to take time off work to participate in screening programs. This aspect of screening could have substantial implications for both individual and societal costs, especially if the screening program does not utilize HPV self-sampling methods, which can mitigate some of these issues.

In light of these considerations, I would recommend expanding your model to include the costs and logistical challenges associated with cervical cancer screening. This would provide a more comprehensive understanding of the cost-effectiveness of the combined vaccination and screening program in Guangdong Province.

Portnoy et al. 2022 highlight the importance of evaluating interventions targeting cervical cancer both independently and in combination. Specifically, their research in Norway examines the interplay between different HPV vaccination strategies (nonavalent versus bivalent) and deintensified screening intervals. This integrated approach provides critical insights into optimizing cervical cancer prevention policies by considering the combined impact of vaccination and screening.

When evaluating the cost-effectiveness of cervical cancer prevention strategies against a benchmark of $40,000 per quality-adjusted life year (QALY) gained, our analysis revealed that maintaining constant screening levels while transitioning Norway's standard vaccination approach from bivalent HPV vaccine (2vHPV) to nonavalent HPV vaccine (9vHPV) would not be economically viable, as indicated by an incremental cost-effectiveness ratio (ICER) of $132,700 per QALY gained. Conversely, our findings suggest that a shift to 9vHPV could be economically justified under a revised threshold of $55,000 per QALY gained, particularly when this shift is combined with a strategy to decrease the total number of screenings over a lifetime.

In light of Portnoy et al.’s findings, I recommend that your analysis could be significantly enhanced by incorporating a similar comprehensive approach. Specifically, it would be valuable to explore how different HPV vaccination strategies, as you have analyzed, interact with varying cervical cancer screening protocols in Guangdong Province. Such an integrated analysis could offer a more nuanced understanding of the overall cost-effectiveness and health outcomes of the proposed interventions.

By considering how vaccination strategies might affect the frequency and nature of screening programs, and vice versa, your study could provide more actionable insights for public health policy makers. This approach would align with the growing recognition in public health research of the need to evaluate primary and secondary prevention efforts in tandem to draw the most accurate and useful conclusions for policy formulation.

Portnoy A, Pedersen K, Nygård M, Trogstad L, Kim JJ, Burger EA. Identifying a Single Optimal Integrated Cervical Cancer Prevention Policy in Norway: A Cost-Effectiveness Analysis. Med Decis Making. 2022 Aug;42(6):795-807. doi: 10.1177/0272989X221082683. Epub 2022 Mar 8. PMID: 35255741.

https://pubmed.ncbi.nlm.nih.gov/35255741/

2. 
In your manuscript, you have focused on modeling the impact of vaccinating schoolgirls against HPV with varying levels of coverage. While your analysis is comprehensive, I have some concerns regarding the feasibility of achieving a 90% vaccination coverage in China. Given the complexities and potential challenges in reaching such high coverage levels, it might be prudent to consider scenarios where the coverage is below 70%.

In situations where girl-only vaccination programs do not achieve high coverage, the efficacy of the program in controlling HPV-related cancers could be limited. To address this issue, it might be beneficial to explore the potential of a gender-neutral vaccination strategy. Such an approach would not only protect boys from HPV-related cancers but also enhance herd protection for girls who are not vaccinated, thereby reducing the overall prevalence of HPV in the population.

Therefore, I would like to suggest the inclusion of a gender-neutral vaccination strategy in your model. Specifically, it would be valuable to analyze the costs and health outcomes associated with extending HPV vaccination to both girls and boys. This addition could provide a more comprehensive understanding of the potential benefits and feasibility of different vaccination strategies in the context of China's public health landscape.

Incorporating the costs and effects of gender-neutral vaccination could offer vital insights into optimizing HPV prevention strategies, especially in scenarios where achieving very high coverage in a girls-only program might be challenging.

3. I've noted in your manuscript the modeling of the transition from the 2v HPV vaccine to the 9v HPV vaccine in your proposed vaccination program. While your focus on the reduction in cervical cancer incidence is commendable, I would like to draw attention to several broader implications of this transition that seem to be absent from your current analysis.

Switching to the 9v HPV vaccine is likely to not only decrease cervical cancer cases but also substantially reduce the overall prevalence of HPV-positive women. This reduction could lead to fewer women requiring follow-up after screening, a decrease in cervical biopsies, and fewer treatments for precancerous lesions. An important consideration here is the impact of these treatments, particularly procedures like LEEP or LLETZ, on subsequent pregnancies. It is well-documented that such treatments can increase the risk of preterm birth, which in turn can lead to significant healthcare costs and emotional burden due to the potential for hospitalization of premature infants.

In light of this, I am curious whether your model includes these broader costs and burdens associated with cervical cancer screening and treatment. Accounting for the downstream effects of reduced HPV prevalence and the need for invasive procedures can provide a more comprehensive view of the cost-effectiveness of the 9v HPV vaccine. This inclusion could potentially reveal additional benefits of the vaccine that extend beyond the direct prevention of cervical cancer, thereby strengthening the case for its implementation in your proposed program.

4. In reviewing your manuscript, I noted the proposed transition in your screening intervention strategy from cervical cytology to HPV DNA-based testing beginning in 2023, with an aim to increase screening coverage to over 70% by 2030. This shift represents a significant change in the approach to cervical cancer screening, which could have profound implications for the effectiveness of the screening program.

However, I find that there is a lack of detailed description of this new HPV DNA-based testing intervention strategy in your manuscript. Specifically, it would be beneficial to have more information on the following aspects:

  1. Start Age of Screening: At what age is the HPV DNA-based screening program proposed to begin? The starting age can have significant implications for the program's overall effectiveness and cost.

  2. Screening Intervals: What are the proposed intervals for HPV DNA-based screening? The frequency of screening is a crucial factor in both the efficacy of the program and its cost-effectiveness.

  3. Differential Strategies for Vaccinated and Unvaccinated Populations: Are there any variations in the screening strategy for cohorts that are HPV-vaccinated versus those that are not? Given the different risk profiles, it might be pertinent to tailor the screening strategies accordingly.

Understanding these elements is vital for comprehensively assessing the impact of the transition to HPV DNA-based testing. The age at which screening starts, the intervals between screenings, and the adaptation of the program for different populations (vaccinated vs. unvaccinated) could significantly influence both the health outcomes and the cost-effectiveness of the program.

Your manuscript would greatly benefit from including this information, as it would provide a clearer understanding of the proposed intervention's structure and its potential implications.

In the study by Pedersen et al. 2018, it was demonstrated that for women vaccinated against HPV, especially with different types of vaccines, the optimal frequency of cervical cancer screening varies. Specifically, the research found that the most cost-effective screening strategy for women vaccinated with the nonavalent (9vHPV) and bivalent/quadrivalent (2/4vHPV) vaccines involves HPV testing once and twice per lifetime, respectively. This indicates that the choice of HPV vaccine has a significant impact on the number of lifetime screenings required to maintain cost-effectiveness in cervical cancer prevention.

Given this insight, I suggest that your model could be further refined by considering how the choice of HPV vaccine (whether 2vHPV, 4vHPV, or 9vHPV) in Guangdong Province might influence the optimal frequency of HPV DNA-based screening. This adjustment would not only align your study with recent findings in the field but also provide a more nuanced understanding of how vaccination strategies can impact screening protocols and overall healthcare costs.

By incorporating these considerations into your analysis, you could offer more tailored and economically efficient recommendations for cervical cancer screening in the context of different HPV vaccination scenarios. This would undoubtedly enhance the practical applicability and relevance of your study's outcomes.

Pedersen K, Burger EA, Nygård M, Kristiansen IS, Kim JJ. Adapting cervical cancer screening for women vaccinated against human papillomavirus infections: The value of stratifying guidelines. Eur J Cancer. 2018 Mar;91:68-75. doi: 10.1016/j.ejca.2017.12.018. Epub 2018 Jan 12. PMID: 29335156; PMCID: PMC5803338.

https://pubmed.ncbi.nlm.nih.gov/29335156/

Minor revisions

Table 3, Upon examining Table 3 in your manuscript, I came across a point that piqued my curiosity and which I believe warrants further explanation. It is noted that with a 90% coverage, the 2-dose 2vHPV vaccine results in 168 cervical cancer cases and 40 deaths, whereas the 2-dose 9vHPV vaccine leads to 160 cases but 41 deaths. This presents an intriguing scenario where a broader-spectrum HPV vaccine (9vHPV) is associated with a slight reduction in the number of cancer cases but an increase in the number of deaths compared to the 2vHPV vaccine.

Could you please provide some insight into how the 9vHPV vaccine, which covers more HPV types and hence presumably offers broader protection, results in a higher mortality rate despite a lower incidence of cervical cancer? This seems counterintuitive as one would expect the broader protection to correspond with not just fewer cancer cases but also fewer deaths.

There are several factors that might contribute to this observation, such as differences in the aggressiveness of the HPV strains covered by the 9vHPV vaccine, variations in the effectiveness of the vaccine against different cancer-causing strains, or potentially even statistical anomalies. However, without a clear explanation, this result could be confusing for readers.

Your clarification on this matter would greatly enhance the understanding of the data presented and the implications of choosing between the 2vHPV and 9vHPV vaccines in terms of their impact on both the incidence and mortality of cervical cancer.

Line 223-225, "Besides, if the universal vaccination program had never been initiated, Guangdong would not be able to achieve cervical cancer elimination by the end of the century, even if the screening target is achieved in 2030."

I would like to inquire about your specific definition of a "universal vaccination program" as mentioned in the manuscript. Does this term refer to a school-based vaccination program, or are you implying a gender-neutral vaccination program that includes both boys and girls? My understanding from your study is that it primarily focuses on vaccination for girls only.

Given the potential differences in impact and feasibility between a school-based program and a gender-neutral one, it is crucial to clarify what is encompassed under the term "universal vaccination program" in the context of your study. This clarification will greatly aid in understanding the assumptions and implications of your model, particularly in terms of the feasibility and effectiveness of the proposed vaccination strategies for cervical cancer elimination in Guangdong.

Line 229-230, "In contrast, HPV vaccination reduces the risk of HPV acquisition in girls before their sexual debut, thereby reducing the risk of cervical cancer later in life." => "In contrast, HPV vaccination is aimed at reducing the risk of HPV acquisition during sexual activity, thereby significantly lowering the incidence of cervical cancer later in life."

The sentence needs to be reformulated because it inaccurately implies that the primary risk of HPV acquisition occurs before sexual debut. In reality, the risk of HPV acquisition is significantly lower before sexual debut, and the primary goal of HPV vaccination is to reduce the risk of acquiring HPV during sexual activity later in life.

This revised sentence correctly emphasizes that the HPV vaccine's primary purpose is to protect against HPV infections that are typically acquired through sexual contact, which is the primary transmission mode of the virus. By vaccinating individuals before they become sexually active, the vaccine can provide immunity against HPV, thereby reducing the risk of developing HPV-related cancers, including cervical cancer, in the future.

Author Response

I would like to take this opportunity to express my sincere gratitude for your time and efforts in reviewing my manuscript. Your positive feedback and encouragement have been instrumental in enhancing the quality of the final manuscript, and I am truly honored to have had the benefit of your expertise and insights.

To facilitate the review process, we have provided a point-by-point response to each of the comments.

Please find below our point-by-point response to your comments:

Comments

  1. Firstly, it appears that your analysis is primarily focused on evaluating the cost-effectiveness of various HPV vaccines, differing vaccine dosages, vaccination coverage levels, and screening coverage rates. While this is undoubtedly important, I noticed that your model does not seem to include the costs associated with the screening program itself, particularly in a vaccinated cohort. The financial implications of screening are critical, especially considering that screening costs can significantly surpass those of HPV vaccination. This is due to the recurring nature of screening throughout an individual's life, in contrast to the one-time or two-time administration of the HPV vaccine.

Additionally, the logistical and economic impacts of screening on adults should be factored into your analysis. Unlike schoolchildren who can be vaccinated during school hours without disrupting work schedules, adults typically need to take time off work to participate in screening programs. This aspect of screening could have substantial implications for both individual and societal costs, especially if the screening program does not utilize HPV self-sampling methods, which can mitigate some of these issues.

In light of these considerations, I would recommend expanding your model to include the costs and logistical challenges associated with cervical cancer screening. This would provide a more comprehensive understanding of the cost-effectiveness of the combined vaccination and screening program in Guangdong Province.

Response: We totally agree with your valuable suggestion. We understand the importance of including the costs and logistical challenges associated with cervical cancer screening. So we have included transportation expenses and charge for loss of working time as indirect costs in the supplementary. Thank you again for your valuable suggestion.

Portnoy et al. 2022 highlight the importance of evaluating interventions targeting cervical cancer both independently and in combination. Specifically, their research in Norway examines the interplay between different HPV vaccination strategies (nonavalent versus bivalent) and deintensified screening intervals. This integrated approach provides critical insights into optimizing cervical cancer prevention policies by considering the combined impact of vaccination and screening.

When evaluating the cost-effectiveness of cervical cancer prevention strategies against a benchmark of $40,000 per quality-adjusted life year (QALY) gained, our analysis revealed that maintaining constant screening levels while transitioning Norway's standard vaccination approach from bivalent HPV vaccine (2vHPV) to nonavalent HPV vaccine (9vHPV) would not be economically viable, as indicated by an incremental cost-effectiveness ratio (ICER) of $132,700 per QALY gained. Conversely, our findings suggest that a shift to 9vHPV could be economically justified under a revised threshold of $55,000 per QALY gained, particularly when this shift is combined with a strategy to decrease the total number of screenings over a lifetime.

In light of Portnoy et al.’s findings, I recommend that your analysis could be significantly enhanced by incorporating a similar comprehensive approach. Specifically, it would be valuable to explore how different HPV vaccination strategies, as you have analyzed, interact with varying cervical cancer screening protocols in Guangdong Province. Such an integrated analysis could offer a more nuanced understanding of the overall cost-effectiveness and health outcomes of the proposed interventions.

By considering how vaccination strategies might affect the frequency and nature of screening programs, and vice versa, your study could provide more actionable insights for public health policy makers. This approach would align with the growing recognition in public health research of the need to evaluate primary and secondary prevention efforts in tandem to draw the most accurate and useful conclusions for policy formulation.

Portnoy A, Pedersen K, Nygård M, Trogstad L, Kim JJ, Burger EA. Identifying a Single Optimal Integrated Cervical Cancer Prevention Policy in Norway: A Cost-Effectiveness Analysis. Med Decis Making. 2022 Aug;42(6):795-807. doi: 10.1177/0272989X221082683. Epub 2022 Mar 8. PMID: 35255741.

https://pubmed.ncbi.nlm.nih.gov/35255741/

Response: Thanks for your valuable suggestion. I have read the article and found that a CEA of a vaccine assuming a very intensive screening strategy may not find the vaccine cost-effective. Reducing screening intensity to make the 9vHPV vaccine more cost-effective compared to the 2vHPV vaccine is interesting. Some scholars in China have also conducted research on controlling the current budget, reducing the intensity of cervical cancer screening, and increasing vaccination coverage so make a move toward an efficient cervical cancer prevention policy. However, the screening policy in China is expected to maintain its intensity for a long time, with the aim of expanding the coverage rate. Considering the continuity of policies, our analysis did not intend to include a reduction in screening intensity as a variable. However, your suggestion is highly insightful, and I will incorporate the consideration of reduced screening into my thesis model analysis. Thank you very much for your input.

Furthermore, I believe it is necessary to explore the impact of reducing screening intensity on vaccine administration within the current budget. Therefore, I have included your suggestion in our discussion as follows: Our study assumes that screening intensity will remain constant in the future, with efforts focused on increasing screening coverage. Under this assumption, the results suggest that the bivalent vaccine is more cost-effective. However, studies in Norway have shown that a reduction in screening intensity makes the 9vHPV vaccine more cost-effective at current prices. It is necessary to include changes in screening intensity in the model to analyze the impact of increased or decreased screening intensity on the cost-effectiveness of the 2vHPV or 9vHPV vaccines. Thank you for your translation request. (lines 367-373)

  1. In your manuscript, you have focused on modeling the impact of vaccinating schoolgirls against HPV with varying levels of coverage. While your analysis is comprehensive, I have some concerns regarding the feasibility of achieving a 90% vaccination coverage in China. Given the complexities and potential challenges in reaching such high coverage levels, it might be prudent to consider scenarios where the coverage is below 70%.

In situations where girl-only vaccination programs do not achieve high coverage, the efficacy of the program in controlling HPV-related cancers could be limited. To address this issue, it might be beneficial to explore the potential of a gender-neutral vaccination strategy. Such an approach would not only protect boys from HPV-related cancers but also enhance herd protection for girls who are not vaccinated, thereby reducing the overall prevalence of HPV in the population.

Therefore, I would like to suggest the inclusion of a gender-neutral vaccination strategy in your model. Specifically, it would be valuable to analyze the costs and health outcomes associated with extending HPV vaccination to both girls and boys. This addition could provide a more comprehensive understanding of the potential benefits and feasibility of different vaccination strategies in the context of China's public health landscape.

Incorporating the costs and effects of gender-neutral vaccination could offer vital insights into optimizing HPV prevention strategies, especially in scenarios where achieving very high coverage in a girls-only program might be challenging.

Response: Thanks for your valuable comments. We totally agree with your suggestion. it might be beneficial to explore the potential of a gender-neutral vaccination strategy. Such an approach would not only protect boys from HPV-related cancers but also enhance herd protection for girls who are not vaccinated, thereby reducing the overall prevalence of HPV in the population. We also know that countries such as the United States, Austria, and Canada which are supported by the Global Alliance for Vaccines and Immunization have been offering HPV vaccine to both genders for several years.

However, because of the lack of local data, our analysis could not examine other diseases caused by HPV, such as oropharyngeal, vagina, head and neck cancers. Meanwhile, according to WHO's global strategy for cervical cancer elimination, the primary target is vaccinating young girls who could directly benefit from it. WHO's Strategic Advisory Group of Experts on Immunisation (SAGE) has recommended that vaccinating boys should be delayed until current vaccine supply constraints are alleviated. Furthermore, there is currently no nationwide policy in China that mandates or recommends HPV vaccination for boys. Therefore, we have added your suggestion to our limitation in the manuscript. We believe that further analysis is necessary. Therefore, we plan to continue our analysis in the future. We appreciate your feedback and will take it into consideration as we further develop our study. (lines 360-362)

  1. I've noted in your manuscript the modeling of the transition from the 2v HPV vaccine to the 9v HPV vaccine in your proposed vaccination program. While your focus on the reduction in cervical cancer incidence is commendable, I would like to draw attention to several broader implications of this transition that seem to be absent from your current analysis.

Switching to the 9v HPV vaccine is likely to not only decrease cervical cancer cases but also substantially reduce the overall prevalence of HPV-positive women. This reduction could lead to fewer women requiring follow-up after screening, a decrease in cervical biopsies, and fewer treatments for precancerous lesions. An important consideration here is the impact of these treatments, particularly procedures like LEEP or LLETZ, on subsequent pregnancies. It is well-documented that such treatments can increase the risk of preterm birth, which in turn can lead to significant healthcare costs and emotional burden due to the potential for hospitalization of premature infants.

In light of this, I am curious whether your model includes these broader costs and burdens associated with cervical cancer screening and treatment. Accounting for the downstream effects of reduced HPV prevalence and the need for invasive procedures can provide a more comprehensive view of the cost-effectiveness of the 9v HPV vaccine. This inclusion could potentially reveal additional benefits of the vaccine that extend beyond the direct prevention of cervical cancer, thereby strengthening the case for its implementation in your proposed program.

Response: We understand the importance of incorporating the broader costs and burdens associated with cervical cancer screening and treatment in order to uncover additional benefits of the vaccine. Our study has included the impact of these treatments, and the benefits have been measured using Disability-Adjusted Life Years (DALYs), which consist of Years of Life Lost (YLL) due to premature mortality in the population, and Years Lost due to Disability (YLD), which includes the reduction in quality of life for women as a result of cervical cancer surgeries.

  1. In reviewing your manuscript, I noted the proposed transition in your screening intervention strategy from cervical cytology to HPV DNA-based testing beginning in 2023, with an aim to increase screening coverage to over 70% by 2030. This shift represents a significant change in the approach to cervical cancer screening, which could have profound implications for the effectiveness of the screening program.

However, I find that there is a lack of detailed description of this new HPV DNA-based testing intervention strategy in your manuscript. Specifically, it would be beneficial to have more information on the following aspects:

Start Age of Screening: At what age is the HPV DNA-based screening program proposed to begin? The starting age can have significant implications for the program's overall effectiveness and cost.

Response: Thanks for your valuable comments. We have carefully added a detailed description of this new HPV DNA-based testing intervention strategy to our manuscript. The target population for cervical cancer screening was women aged 35–64 years. We assumed that the start age of screening was done at the age of 35 years because the highest prevalence of cervical precancerous lesions and undetected cervical cancers could be detected at this age. We appreciate your feedback and have taken it into consideration. (lines 143-153)

Table 3, Upon examining Table 3 in your manuscript, I came across a point that piqued my curiosity and which I believe warrants further explanation. It is noted that with a 90% coverage, the 2-dose 2vHPV vaccine results in 168 cervical cancer cases and 40 deaths, whereas the 2-dose 9vHPV vaccine leads to 160 cases but 41 deaths. This presents an intriguing scenario where a broader-spectrum HPV vaccine (9vHPV) is associated with a slight reduction in the number of cancer cases but an increase in the number of deaths compared to the 2vHPV vaccine.

Could you please provide some insight into how the 9vHPV vaccine, which covers more HPV types and hence presumably offers broader protection, results in a higher mortality rate despite a lower incidence of cervical cancer? This seems counterintuitive as one would expect the broader protection to correspond with not just fewer cancer cases but also fewer deaths.

There are several factors that might contribute to this observation, such as differences in the aggressiveness of the HPV strains covered by the 9vHPV vaccine, variations in the effectiveness of the vaccine against different cancer-causing strains, or potentially even statistical anomalies. However, without a clear explanation, this result could be confusing for readers.

Your clarification on this matter would greatly enhance the understanding of the data presented and the implications of choosing between the 2vHPV and 9vHPV vaccines in terms of their impact on both the incidence and mortality of cervical cancer.

Response: Thanks for your valuable comments. We apologize for making a mistake in the original manuscript. We have revised it.

Line 223-225, "Besides, if the universal vaccination program had never been initiated, Guangdong would not be able to achieve cervical cancer elimination by the end of the century, even if the screening target is achieved in 2030."

I would like to inquire about your specific definition of a "universal vaccination program" as mentioned in the manuscript. Does this term refer to a school-based vaccination program, or are you implying a gender-neutral vaccination program that includes both boys and girls? My understanding from your study is that it primarily focuses on vaccination for girls only.

Given the potential differences in impact and feasibility between a school-based program and a gender-neutral one, it is crucial to clarify what is encompassed under the term "universal vaccination program" in the context of your study. This clarification will greatly aid in understanding the assumptions and implications of your model, particularly in terms of the feasibility and effectiveness of the proposed vaccination strategies for cervical cancer elimination in Guangdong.

Response: We truly agree with your suggestion. The term “universal vaccination program" as mentioned in the manuscript refers to a school-based vaccination program for girls. Guangdong province has included the vaccination of adolescent girls with two doses of the domestic HPV vaccine before their sexual debut in 2023. Thus, the target of vaccination is schoolgirls. We have clarified the term "universal vaccination program" in the context of the study. And revised the sentence as "Besides, if the universal vaccination program for schoolgirls had never been initiated, Guangdong would not be able to achieve cervical cancer elimination by the end of the century, even if the screening target is achieved in 2030." (lines 59-60, lines 313)

Line 229-230, "In contrast, HPV vaccination reduces the risk of HPV acquisition in girls before their sexual debut, thereby reducing the risk of cervical cancer later in life." => "In contrast, HPV vaccination is aimed at reducing the risk of HPV acquisition during sexual activity, thereby significantly lowering the incidence of cervical cancer later in life."

The sentence needs to be reformulated because it inaccurately implies that the primary risk of HPV acquisition occurs before sexual debut. In reality, the risk of HPV acquisition is significantly lower before sexual debut, and the primary goal of HPV vaccination is to reduce the risk of acquiring HPV during sexual activity later in life.

This revised sentence correctly emphasizes that the HPV vaccine's primary purpose is to protect against HPV infections that are typically acquired through sexual contact, which is the primary transmission mode of the virus. By vaccinating individuals before they become sexually active, the vaccine can provide immunity against HPV, thereby reducing the risk of developing HPV-related cancers, including cervical cancer, in the future.

Response: We truly agree with your suggestion. We have revised the sentence. (lines 309-310)

We tried our best to improve our manuscript and made some changes in the manuscript. We appreciate for the reviewer’s warm work earnestly, and hope that the correction will meet with approval. Once again, thank you very much for your comments and suggestions.

Looking forward to hearing from you soon.

Yours Sincerely,

               on behalf of all authors.

Round 2

Reviewer 1 Report

Comments and Suggestions for Authors

I am satisfied with the modifications.